

# Effect of floods on the $\delta^{13}C$ values in plant leaves: a study of willows in Northeastern Siberia

Rong Fan[1], Tomoki Morozumi[1], Trofim C. Maximov[2,3] and Atsuko Sugimoto[3,4,5,6]

[1] Graduate School of Environmental Science, Hokkaido University, Sapporo, Hokkaido, Japan
[2] Institute for Biological Problems of Cryolithozone, Siberian Blanch of Russian Academy of Sciences, Yakutsk, Sakha, Russia
[3] North Eastern Federal University, Yakutsk, Sakha, Russia
[4] Arctic Research Center, Hokkaido University, Sapporo, Hokkaido, Japan
[5] Global Station for Arctic Research, Global Institution for Collaborative Research and Education, Hokkaido University, Sapporo, Hokkaido, Japan
[6] Faculty of Environmental Earth Science, Hokkaido University, Sapporo, Hokkaido, Japan

Corresponding authors
Rong Fan,
fanrong@ees.hokudai.ac.jp
Atsuko Sugimoto,
atsukos@ees.hokudai.ac.jp

## ABSTRACT

Although stable carbon isotopic composition ($\delta^{13}C$) of plants has been widely used to indicate different water regimes in terrestrial ecosystems over the past four decades, the changes in the plant $\delta^{13}C$ value under waterlogging have not been sufficiently clarified. With the enhanced global warming in recent years, the increasing frequency and severity of river floods in Arctic regions lead to more waterlogging on willows that are widely distributed in river lowland. To investigate the $\delta^{13}C$ changes in plants under different water conditions (including waterlogging), we measured the $\delta^{13}C$ values in the leaves of willows with three species, *Salix boganidensis*, *S. glauca*, and *S. pulchra*, and also monitored changes in plant physiology, under several major flooding conditions in Northeastern Siberia. The foliar $\delta^{13}C$ values of willows varied, ranging from −31.6 to −25.7‰ under the different hydrological status, which can be explained by: (i) under normal conditions, the foliar $\delta^{13}C$ values decrease from dry (far from a river) to wet (along a river bank) areas; (ii) the $\delta^{13}C$ values increase in frequently waterlogged areas owing to stomatal closure; and (iii) after prolonged flooding periods, the $\delta^{13}C$ values again decrease, probably owing to the effects of not only the closure of stomata but also the reduction of foliar photosynthetic ability under long period of waterlogging. Based on these results, we predict that plant $\delta^{13}C$ values are strongly influenced by plant physiological responses to diverse hydrological conditions, particularly the long periods of flooding, as occurs in Arctic regions.

## INTRODUCTION

Over the past four decades, stable carbon isotopic composition ($\delta^{13}C$, ‰ relative to Vienna Pee Dee Belemnite, VPDB) of plants has been widely employed as a conventional tool to estimate changes in carbon flux as plant physiology responds to environmental

changes. The magnitude of isotopic fractionation is indeed highly dependent on physiological conditions (*Farquhar, Ehleringer & Hubick, 1989*; *Robinson, 2001*). For instance, it is well known that carbon isotopic fractionation ($\Delta^{13}C$) in plants is a function of the ratio of leaf intercellular-($c_i$) to atmospheric ($c_a$) $CO_2$ concentrations ($c_i/c_a$) (*Farquhar & Sharkey, 1982*; *Farquhar, Ehleringer & Hubick, 1989*), as given in Eq. (1):

$$\Delta^{13}C = \delta^{13}C_a - \delta^{13}C_p = a + (b - a) \times \frac{C_i}{C_a}, \tag{1}$$

where, $\delta^{13}C_a$ and $\delta^{13}C_p$ are the $\delta^{13}C$ values of atmospheric $CO_2$ and photosynthate, respectively; while $a$ and $b$ are the carbon isotopic fractionations associated with $CO_2$ diffusion and enzymatic carboxylation (carbon fixation) in plant leaves, respectively.

The $c_i/c_a$ ratio is usually determined from the balance between the $CO_2$ supply controlled by stomatal conductance and $CO_2$ consumption via the carboxylation related to photosynthetic activity. When the stomata closes (e.g., in response to a large water deficit and high evaporation rates due to high ambient temperature, (*Meidner & Mansfield, 1968*; *Willmer & Fricker, 1996*)), low $CO_2$ supply reduces $c_i$, leading to a decrease in the $\Delta^{13}C$ values and, ultimately, an increase in the $\delta^{13}C_p$ values. On the other hand, when $CO_2$ consumption decreases as a consequence of reducing photosynthetic activity (e.g., due to the limitations of light and nutrients (*Hall & Krishna, 1999*)), a large $c_i$ increases the $\Delta^{13}C$ values and, ultimately, decreases the $\delta^{13}C_p$ values. These effects are expressed by Eq. (2),

$$\frac{A}{gc} = C_a - C_i = C_a \times \left(1 - \frac{C_i}{C_a}\right), \tag{2}$$

where $A$ is the photosynthetic rate, $gc$ is the stomatal conductance of $CO_2$, and $gs$ is stomatal conductance which equals 1.6 times $gc$.

Combining Eqs. (1) and (2), the $\Delta^{13}C$ and $\delta^{13}C_p$ values are given by the standard Eq. (3):

$$\Delta^{13}C = \delta^{13}C_a - \delta^{13}C_p = b - \frac{b - a}{C_a} \times \frac{1.6A}{gs}, \tag{3}$$

Thus, the plant $\delta^{13}C$ values are primarily controlled by both stomatal conductance ($gs$) for $CO_2$ and photosynthetic activity ($A$) (*Farquhar & Richards, 1984*). For example, under constant $A$, the $\delta^{13}C_p$ values are controlled mainly by the $gs$. Drought-induced low $gs$ decreases the $\Delta^{13}C$ values and increases the $\delta^{13}C_p$ values. In contrast, moisture-induced high $gs$ increases the $\Delta^{13}C$ values and decreases the $\delta^{13}C_p$ values (*Farquhar & Richards, 1984*; *Knight, Livingston & Van Kessel, 1994*; *Korol et al., 1999*; *Barbour & Farquhar, 2000*; *Warren, McGrath & Adams, 2001*; *Huang et al., 2008*; *Peri et al., 2012*). Under constant $gs$, however, the $\delta^{13}C_p$ values are primarily controlled by $A$, which is strongly correlated with light intensity (*Yakir & Israeli, 1995*) and nutrient availability (*Ripullone et al., 2004*; *Duursma & Marshall, 2006*; *Kranabetter et al., 2010*). Enhanced $A$ decreases the $\Delta^{13}C$ values and, ultimately, increases the $\delta^{13}C_p$ values (*O'Leary, 1988*; *Farquhar, Ehleringer & Hubick, 1989*).

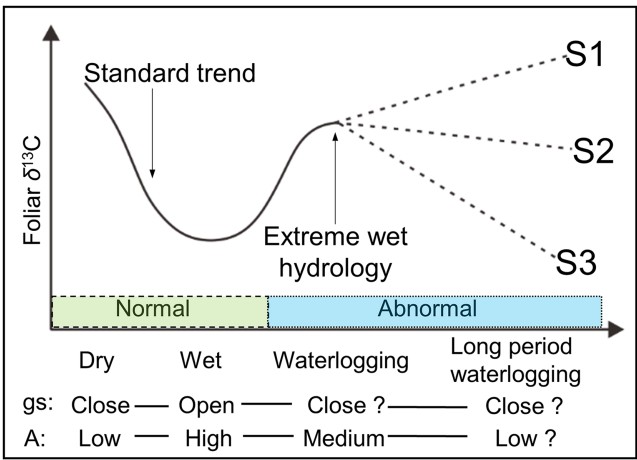

**Figure 1 Schematic view of the possible foliar δ¹³C values under various hydrological conditions.** $gs$ and $A$ are stomatal conductance and photosynthetic activity, respectively. Dry and wet are without waterlogging, and waterlogging and long period waterlogging represent continual and continuous waterlogging, respectively. Possible changes in the foliar δ¹³C value are shown for assumed scenarios (S1, S2, and S3).

With respect to the river flooding, there is physiological evidence that stomata can also be closed in response to waterlogging conditions (*Gomes & Kozlowski, 1980*; *Olivella et al., 2000*; *Copolovici & Niinemets, 2010*); though, such evidence does not include isotope data, such as the $\Delta^{13}C$ or $\delta^{13}C_p$ values. If the $CO_2$ $gs$ term in Eq. (3) decreases due to the low stomatal conductance during waterlogging, low foliar $\Delta^{13}C$ (high foliar $\delta^{13}C_p$) values will appear very similar to the values observed under drought conditions. Indeed, previous studies have reported changes in the $\delta^{13}C_p$ value under both, natural and simulated waterlogging conditions. *Anderson et al. (2005)* found that tree-ring $\delta^{13}C$ values for the pond cypress *Taxodium ascendens* in their natural environments are positively correlated with the total annual precipitation; similarly, *Li & Sugimoto (2017)* reported an increase in needle $\delta^{13}C_p$ values for the larch *Larix gmelinii* in waterlogging pot experiments. The latter study attributed the increase in larch needle $\delta^{13}C_p$ values to low $gs$ caused by waterlogging.

Although *Anderson et al. (2005)* reported a decreased $gs$ with an increased $A$ under very wet conditions, most physiological experiments have demonstrated that not only $gs$, but also $A$ is apparently reduced under waterlogging (*Gomes & Kozlowski, 1980*; *Copolovici & Niinemets, 2010*; *Li & Sugimoto, 2017*). Based on these findings, we hypothesize that the $\Delta^{13}C$ (and $\delta^{13}C_p$) values in plant leaves are not exclusively under the controlling of stomata ($gs$), because the photosynthetic rate ($A$) is also not constant in waterlogging. Moreover, net $A$ and chlorophyll contents were observed decreasing without any change in either $gs$ or $c_i/c_a$, in a continuous waterlogging experiment with okra *Abelmoschus esculentus* (*Ashraf & Arfan, 2005*), a waterlogging-tolerant plant. Thus, the possible changes in the foliar δ¹³C value under long period (continuous) waterlogging, which are assumed more dependent on changes of photosynthetic rate, are shown in Fig. 1 under predicted scenarios 1, 2, and 3 (S1, S2, and S3). Moreover, it is possible, as S3, that under low $gs$, a reduction of $A$ can lead to lower $\delta^{13}C_p$ values. Thus, $\Delta^{13}C$ (and $\delta^{13}C_p$)

values will be potentially changed in terms of frequency, magnitude, and duration of waterlogging.

The Arctic region is highly sensitive and responsive to climatic changes (*Giorgi, 2006*). Thus, increases in atmospheric temperature significantly affect the hydrology in this region, including prevailing spring floods (*Shahgedanova, 2002*; *Shiklomanov et al., 2007*; *Tan, Adam & Lettenmaier, 2011*). For example, with rising temperature, the annual average discharge rate from the 19 largest rivers in the Arctic increased by approximately 10% from 1977 to 2007 (*Overeem & Syvitski, 2010*). Since the topography of the Arctic river lowlands is relatively flat, spring flooding strongly influences riparian plant communities. Shrubs which can stand high moisture levels, predominate over low moisture-preferring trees like larch and pine in areas along rivers under recurrent spring floods (*Troeva et al., 2010*). For instance, in the wide Indigirka River lowland near Chokurdakh village Russia in Northeastern Siberia, one sixth of a $10 \times 10$ km$^2$ area is covered by dwarf shrub willow (*Salix*) (T. Morozumi, 2015, personal communication) and particularly being abundant on river banks. Thus, because willows in this area are exposed to an increase frequency of river floods and have high chances to be submerged, they are a good candidate species to study the effects of flooding on the $\delta^{13}C_p$ values of leaves in relation to $A$ and $gs$.

The objective of this study was to determine the effects of flooding on the $\delta^{13}C$ values in willow leaves under four major hydrological conditions: dry, wet, and short and long period waterlogging (Fig. 1). We measured the $\delta^{13}C$ values of bulk leaves from willows growing under these flooding regimes in the Indigirka River lowland of Northeastern Siberia.

## MATERIALS AND METHODS

### Study area

The study site is located in the Indigirka River lowland near Chokurdakh (70°38′N, 147°53′E), Sakha Republic (Yakutia), Russian Federation (Fig. 2). Mean annual air temperature in the region between 1950 and 2016 was −13.7 °C, ranging from −33.9 °C in January (the coldest month) to 10.1 °C in July (the warmest month). Mean annual precipitation between 1950 and 2008 was 209 mm year$^{-1}$ (*Yabuki et al., 2011*).
The Indigirka River lowland, including rivers, lakes, wetlands, hills, and floodplains, is frequently flooded during spring and summer. Soils in the region are loamy or silty-loamy alluvial soils with black- to grayish-olive color along the riverbanks (*Troeva et al., 2010*). The average depth of the active layer in soils is approximately 30 cm on land and one m near the river in the summer. The local vegetation consists of aquatics, sphagnums mosses, graminoids, shrubs (mainly the willow *Salix sp.* and the dwarf birch *Betula nana*), alders, larches, and pines. Between 1970 and 2016, the average intra-annual water level cycle of the Indigirka River was 70 ± 83 mm for April and May (late winter, pre-flooding), increasing to 600 ± 93 mm for June–August (spring and summer, flooding season); then, gradually receding to 343 ± 146 mm for September and October (autumn and early winter, post-flooding), and declining further to 56 ± 26 mm in winter (after October). Field experiments were approved by Hokkaido University, and Institute for Biological

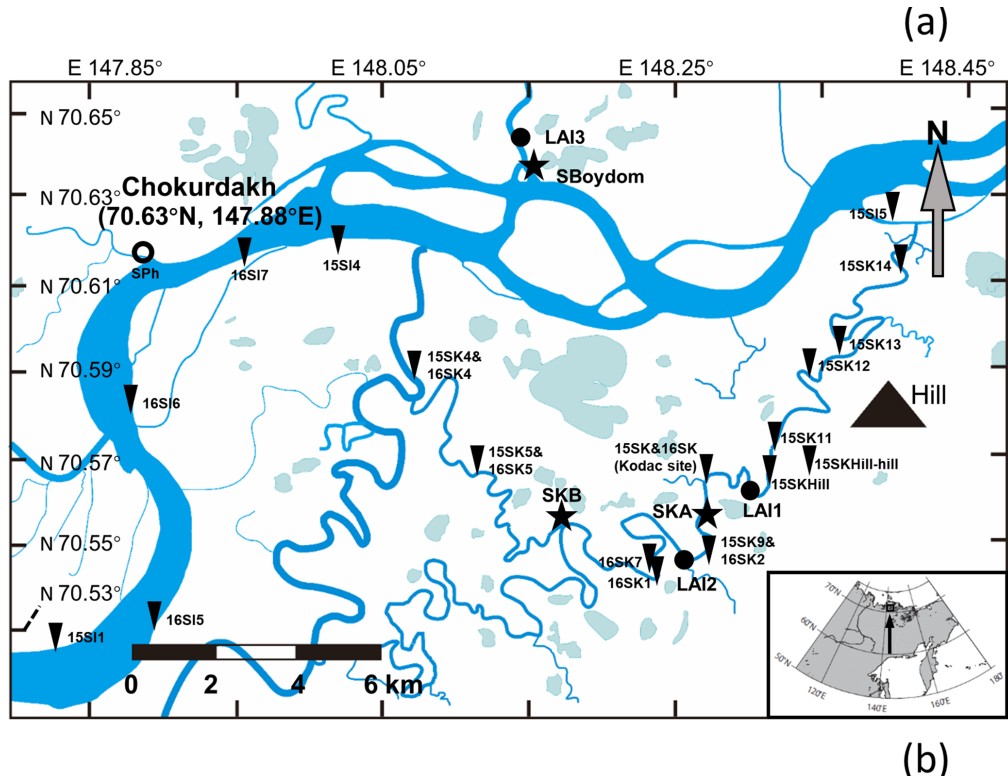

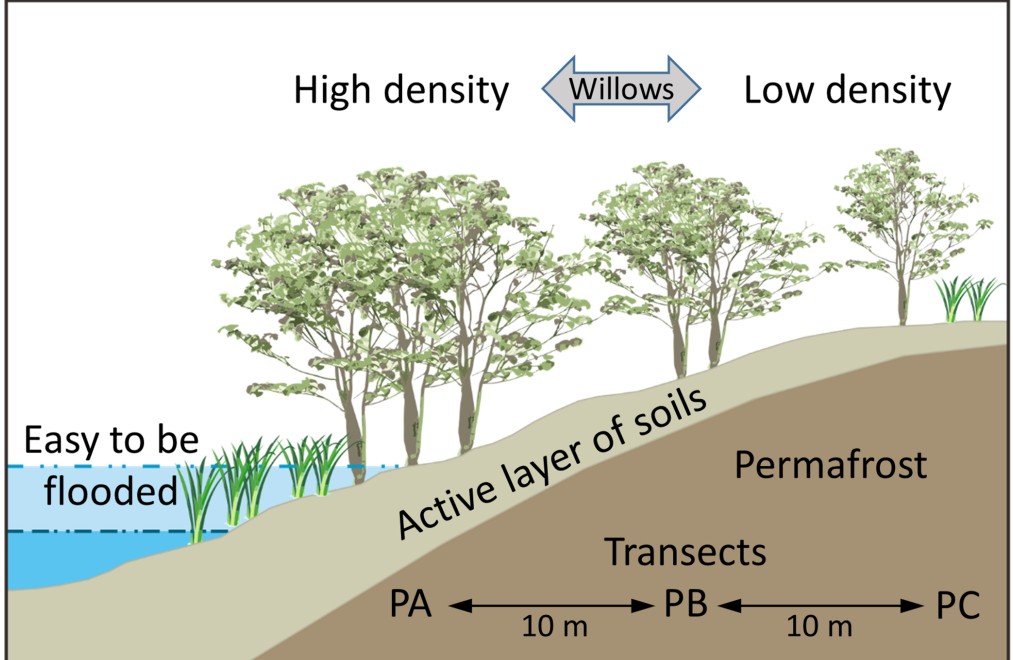

Figure 2 **Sampling sites and schematic illustration of a transect.** (A) Sampling sites near Chokurdakh village in the study region, northeastern Siberia. Thick and thin blue lines represent the Indigirka River and its tributaries, respectively. Areas filled with light blue represent lakes. Triangles (18), stars (3), filled black circles (3) and empty circle (1) indicate the sampling sites, three transects (SKA, SKB, and SBoydom), three sites for production measurement (LAI1~3) and one site for photosynthesis monitoring (SPh). More sampling sites see Table A2. (B) A schematic illustration of a transect.

Problems of Cryolithozone, Siberian Branch of Russian Academy of Science, and North-Eastern Federal University.

## Willows in the Indigirka River lowland

The common willow species observed in 2015–2017 were *Salix boganidensis*, *S. pulchra*, *S. glauca*, *S. richardsonii*, *S. viminalis*, *S. alaxensis*, *S. fuscescens*, and *S. hastata*. Most species were ≤1 m tall, except for a few species such as *S. boganidensis*, *S. alaxensis*, and *S. fuscescens*, which were two to three m in height. Diameter at breast height generally ranged between one and six cm. Maximum root depth was approximately one m at the riverbank, but was highly variable and depended on various factors such as the thickness of the active soil layers and moisture levels where the willows grew. Willows were distributed more densely along the riverbanks than on dry lands.

Observations conducted with a GardenWatchCam time-lapse camera (Brinno, Inc., Taipei City, Taiwan) showed that the buds of willow leaves opened around the first few weeks of June, when the snow had melted and the daily average air temperature had increased to >0 °C. The leaves and stems grew rapidly, within 10 days after bud opening, and were fully developed by mid-July. Willow leaf biomass peaked by the end of July, and this observation was consistent with a normalized difference vegetation index (NDVI) study in Alaska (*Boelman et al., 2011*). Aboveground net primary production (ANPP, newly formed stems and leaves in each year) and the leaf area index (LAI) of the willows in 2016 were measured using the direct harvesting method (*Jonckheere et al., 2004*) in three blocks which were predominated by willows. ANPP was 63, 119, and 117 g m$^{-2}$·a in each of the three blocks, and the LAI was 0.59, 0.71, and 1.59 in each of the three blocks (Table A1).

## Samples

In the summer of 2015 and 2016, we collected leaves from the locally dominant willows *Salix boganidensis*, *S. glauca*, and *S. richardsonii* on three sets of 20 m transects (SBoydom, SKA, and SKB) from the river. SBoydom is located between the mainstream Indigirka and the wetland; while, SKA and SKB are situated next to a secondary tributary, Kryvaya (Fig. 2; Table A2). Three points, named PA, PB, and PC, were marked on each transect based on their distance to the river. The maximum thaw depth was always found at PA. This layout was designed based on the differences in intra- and inter-annual flooding conditions (Figs. 2 and 3). PAs at SKB and SBoydom were continually waterlogged throughout the growing season in 2015 and continuously waterlogged until July in 2016 (Fig. 3). PB at SKB and PA at SKA were flooded only in 2016 (Fig. 3).

Four current-year top shoots were collected at each point at the end of the growing season (the end of July) in both 2015 and 2016. Current-year shoots were also randomly sampled from willows in local scale on the Indigirka River lowland during the same period. A total of 31 sites with different locations were used in 2015 and 2016 (Fig. 2; Table A2). At least four current-year shoots were collected at each location to obtain representative data for each site.

The details of sampling sites, locations, species, and sampling numbers are shown in Fig. 2 and Table A2. All samples were immediately dried at 60 °C for 48 h after collecting.

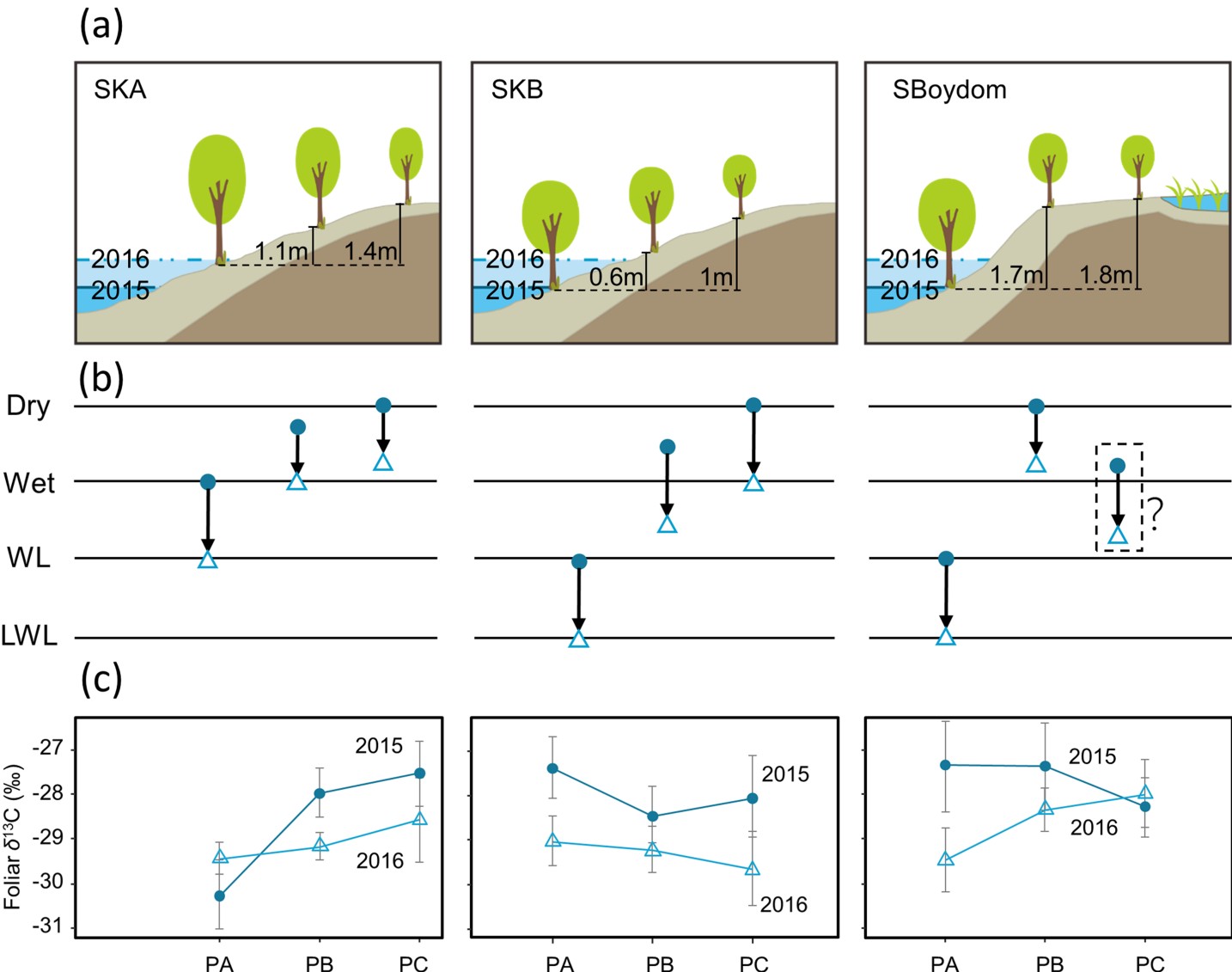

**Figure 3 Schematic view of each transect, possible changes in the hydrological conditions, and foliar δ¹³C values (‰).** (A) Schematic view of each transect with the highest water levels observed in each case in 2015 (blue) and 2016 (light blue) and the height of PB and PC, compared to PA (black line). (B) Possible changes (black arrows) in the hydrological conditions from 2015 (filled circles) to 2016 (open triangles) in each point on transects. Dry and wet are without waterlogging, and WL and LWL represent waterlogging (continual) and long period waterlogging (continuous), respectively. (C) The foliar δ¹³C values (‰) found in willows were reported as mean ± SD.

### Stable carbon isotope analysis

Dried leaves were milled into fine powder with liquid $N_2$ and dried again at 60 °C for 48 h; each sample was then wrapped in a tin capsule and injected into an elemental analyzer (Flash EA 1112; Thermo Fisher Scientific, Bremen, Germany), connected to an isotope ratio mass spectrometry (IRMS, Delta V; Thermo Fisher Scientific, Bremen, Germany) through a continuous-flow carrier-gas system (Conflo III; Thermo Fisher Scientific, Bremen, Germany). The stable carbon isotopic composition was reported in the standard δ notation relative to VPDB. A laboratory standard was injected after every ten samples

to verify that the analytical accuracy was better than 0.1‰. To reduce the effect of sampling heterogeneity in $\delta^{13}$C within a single site, four samples were measured and the average isotopic composition was reported for each site, with the standard deviation ranging from 0.0 to 1.4‰ (average, 0.7‰).

**Photosynthetic rate and stomatal conductance analyses**

Supporting data on the foliar $\delta^{13}$C values, the photosynthetic rate and stomatal conductance of willow leaves were monitored in the field in 2017 using a portable porometer (LCpro+; ADC BioScientific Ltd, Hoddesdon, Herts, UK) equipped with a conifer chamber and a lighting system. The photosynthetic rate ($A$) of *S. boganidensis*, *S. richardsonii*, and *S. glauca* under different light levels (10–955 µmol m$^{-2}$ s$^{-1}$) was measured to obtain light response curves and thus, to identify the saturation light intensity.

Site SPh near Chokurdakh village, was set up in the summer of 2017 to monitor the conditions in former transects SKA, SKB, and SBoydom, since the extremely high flooding caused all these three sites totally submerged for the entire summer of 2017. Under gradient flooding conditions on site SPh (-PA: submerged till July 20; -PB: submerged till July 15; -PC: without submergence during the observation period), temporary changes were measured in the photosynthetic rate ($A$) and stomatal conductance ($gs$) of *S. richardsonii*, *S. glauca*, and *S. boganidensis* in response to a single saturated light exposure at 600 µmol m$^{-2}$ s$^{-1}$ around noon, the rest of chamber conditions were set to match ambient conditions. For each measurement at the points (PA, PB, or PC), a total of 12 leaves from four trees were marked for leaf ADC data recording for more than six times on any leaf of them. Average of all records was calculated for each measurement. Measurements were taken five times every 2–3 days between July 13, 2017 and July 27, 2017. The leaves were also collected after whole monitoring period to check the foliar $\delta^{13}$C values.

**Statistical analysis**

Linear Mixed Models (LMMs) were used to clarify differences in the foliar $\delta^{13}$C value among willows growing in three transects in 2015 and 2016. Foliar $\delta^{13}$C value was set as the response variable, with flooding condition was set as the fixed effect, and species (i.e., *S. boganidensis*, *S. richardsonii*, and *S. glauca*) was set as a random effect. Similar analyses by LMMs were also used to figure out any differences in the foliar $\delta^{13}$C value among the willows randomly collected on the Indigirka River lowland in 2015 and 2016. Foliar $\delta^{13}$C value was set as the response variable, the flooding condition was assigned as the fixed effect, and the location (along the mainstream or the tributary), and species (Table A2), were set as random effects. Tukey's test was used as a post hoc analysis for multiple comparisons. The lme4 package (*Bates, Maechler & Walker, 2015*) of R (*R Core Team, 2015*) was used to build the LMMs.

# RESULTS

## Foliar $\delta^{13}$C in the transects

Foliar $\delta^{13}$C values differed among SKA, SKB, and SBoydom, and between years (Fig. 3). Along transect SKA in 2015, the mean foliar $\delta^{13}$C values were −30.3 ± 0.8‰ for PA

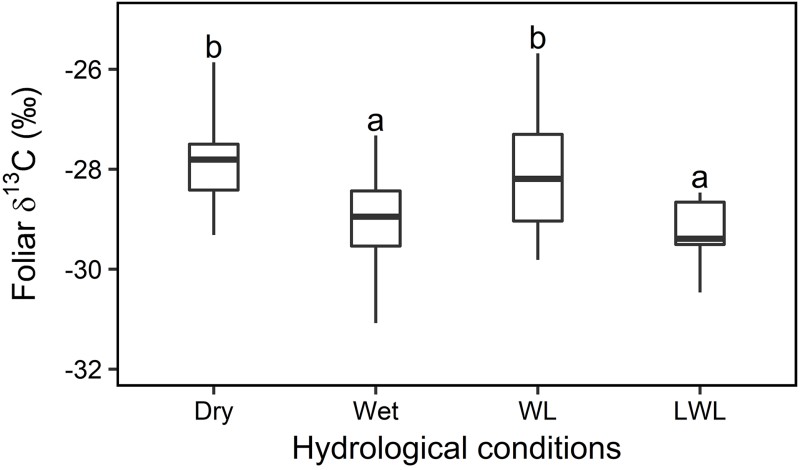

**Figure 4 Statistical analysis for the foliar $\delta^{13}$C values (‰) under four different hydrological conditions in transects in 2015 and 2016.** Box-and-whisker plot of the statistical analysis for the foliar $\delta^{13}$C values (‰) under four different hydrological conditions in sampling transects established in year 2015 and 2016. Different letters over the numbers indicate statistically significant differences according to Turkey's post hoc test and Linear Mixed Model. Dry and wet are without waterlogging, and WL and LWL represent waterlogging (continual) and long period waterlogging (continuous), respectively.

(close to the river but without submergence), which was much lower than those for PB ($-28.0 \pm 0.6$‰) and PC ($-27.5 \pm 0.7$‰). A similar trend was also observed in 2016, when the water level was high; in this case, the foliar $\delta^{13}$C values for PA ($-29.4 \pm 0.4$‰) was lower than those for PB ($-29.2 \pm 0.3$‰) and PC ($-28.6 \pm 0.9$‰); although, the difference was small. For the inter-annual changes from 2015 to 2016 on transect SKA, an increase in foliar $\delta^{13}$C value was observed for PA ($+0.8 \pm 0.6$‰); whereas, a decrease was recorded for both, PB ($-1.2 \pm 0.5$‰) and PC ($-1.0 \pm 0.8$‰).

In 2015, the sampling point at PAs along both, the SKB and SBoydom transects, were sometimes waterlogged ("WL"; Fig. 3); whereas, all points PBs and PCs were not. A similar trend for the foliar $\delta^{13}$C values was found in both, SKB and SBoydom transects, as the foliar $\delta^{13}$C values for PAs ($-27.4 \pm 1.1$‰ and $-27.3 \pm 0.7$‰, respectively) were higher than those for PBs ($-28.5 \pm 0.6$‰ and $-27.4 \pm 1.0$‰, respectively), and PCs ($-28.1 \pm 0.9$‰ and $-28.3 \pm 0.7$‰, respectively) (Fig. 3). In 2016, although PAs were also but always waterlogged ("LWL"; Fig. 3), PBs and PCs were not, trends in the foliar $\delta^{13}$C value were apparently different between SKB and SBoydom. In SKB, the foliar $\delta^{13}$C values for PA ($-29.0 \pm 0.6$‰) were slightly higher than those for PB ($-29.2 \pm 0.5$‰) and PC ($-29.7 \pm 0.9$‰); whereas, in SBoydom, values for PA ($-29.5 \pm 0.7$‰) were slightly lower than those for PB ($-28.3 \pm 0.5$‰) and PC ($-28.0 \pm 0.8$‰) (Fig. 3). For the inter-annual changes in SKB and SBoydom, decreases in the foliar $\delta^{13}$C value were observed at all points in all transects, except for PC in SBoydom. The differences in the foliar $\delta^{13}$C value between the PB and PC within each transect ranging from $-0.5$ to $+0.9$‰; however, those for the PA significantly deviated ($-2.5$ to $+0.9$‰) from the mean value of the PB and PC.

Overall, statistical analysis of the foliar $\delta^{13}$C values from transects (in Fig. 3) showed a significant difference ($F_{3,42} = 42.276$, $P < 0.01$, Fig. 4) among the four major

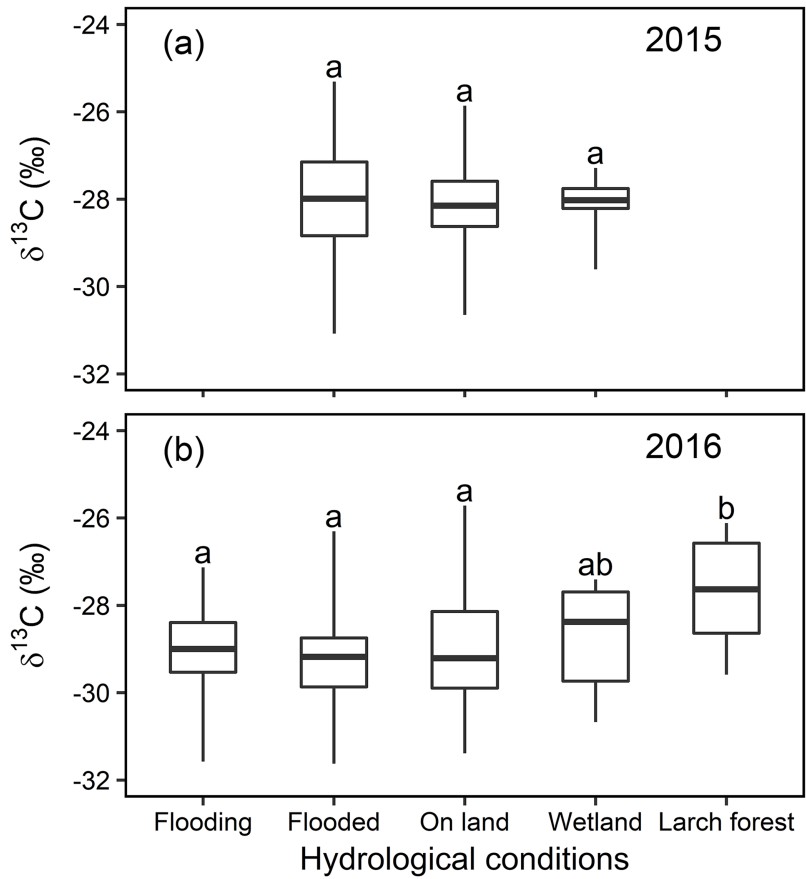

**Figure 5** **Statistical analysis for the foliar δ¹³C values (‰) under different hydrological conditions at local scale in 2015 and 2016.** Box-and-whisker plot of the statistical analysis for the foliar $\delta^{13}C$ values (‰) under different environments with distinct hydrological conditions at local scale in year 2015 (A) and 2016 (B), different letters over the numbers indicate statistically significant differences according to Tukey's post hoc test and the Linear Mixed Model.

hydrological conditions (i.e., "Dry," "Wet," "WL," and "LWL," in Figs. 3 and 4). The foliar $\delta^{13}C$ values were high in dry and waterlogged continually conditions. Conversely, values under wet and continuous long period waterlogging were consistently low.

## Spatial distribution in the foliar δ¹³C of willows

Neither significant nor large differences were detected in the foliar $\delta^{13}C$ value in any of the willows growing in the 31 randomly selected sampling sites in local scale (with different locations) in Indigirka River lowland during the same sampling periods at the end of the growing seasons of 2015 and 2016 (Fig. 5). The willow foliar $\delta^{13}C$ values ranged from −31.1 to −25.3‰ in 2015, and from −31.6 to −25.7‰ in 2016. Statistical analysis shows that there were no significant differences between the four environmental conditions (flooding, flooded, on land, and wetland) in 2015 or 2016. Moreover, the $\delta^{13}C$ values measured for willow leaves collected in a larch forest in 2016 (−27.6 ± 1.2‰) were significantly higher than those sampled anywhere else in the same year ($F_{4,168} = 2.58$, $P = 0.039$).

## Photosynthetic rate and stomatal conductance

Photosynthetic rate ($A$) gradually increased asymptotically and reached to about six µmol m$^{-2}$ s$^{-1}$ for *S. richardsonii*, about six to eight µmol m$^{-2}$ s$^{-1}$ for *S. glauca*, and about 8–12 µmol m$^{-2}$ s$^{-1}$ for *S. boganidensis*, under 400–600 µmol m$^{-2}$ s$^{-1}$ (irradiation scanning covered the range from 10 to 955 µmol m$^{-2}$ s$^{-1}$) (Fig. A1). Therefore, the saturating light intensity for willow leaves in the Indigirka River lowland was found in the range from 400 to 600 µmol m$^{-2}$ s$^{-1}$. Maximum photosynthetic rate $A$ recorded in leaves of *S. boganidensis* was the highest among all three species (Fig. A1).

In the summer of 2017, willows at the SPh-PA were continuously submerged until July 20; willows at the SPh-PB had just come out of the water when monitoring began on July 15; while, willows at the SPh-PC were not submerged during the monitoring period (Fig. 6A), although all three points at transect SPh were within 20 m from the river. The largest decrease (−1.6 ± 1.4 µmol m$^{-2}$ s$^{-1}$) in $A$ at SPh-PA during the monitoring period, was registered on July 21 when the waterlogging just finished; whereas, after flooding $A$ was observed to follow a slow recovery over the last few days (Fig. 6B). A similar decrease-increase trend in $A$ was also detected in the willow leaves at SPh-PB, and also, in this case, the lowest $A$ was recorded soon after waterlogging finished on July 18 (−0.7 ± 2.6 µmol m$^{-2}$ s$^{-1}$ lower than first measurement) (Fig. 6B). However, at SPh-PC, $A$ continuously increased, compared to the initial measurement. These values corresponded with the waterlogging gradients in SPh, that $A$ was reduced under waterlogging and could recover if waterlogging was over. On the other hand, among points, compared to SPh-PC, the lowest $gs$ values were found at SPh-PA; while, intermediate values were recorded in SPh-PB (Fig. 6C). Thus, $gs$, apparently correlated to the degree of waterlogging among SPh-PA, -PB, and -PC.

## DISCUSSION

### River water level and leaf formation

As mentioned before, the willow leaves began opening after the first week in June and finished growing by the end of July. This suggests that the foliar δ$^{13}$C values of willow leaves recorded hydrological conditions experienced from early mid-June to the date of collection, which is a longer period than that of the in situ observation and monitoring period for hydrological conditions. However, it is known that the river water level is gradually decreased but by within approximately one m during the term of leaf growing (Fig. A2). Moreover, only small differences were found in the foliar δ$^{13}$C values between top and bottom of a single current-year shoot, approximately 0.5 ± 0.1‰ (Fig. A3), which may have experienced leaf formation over different periods. Thus, in present field observation study, the hydrological conditions observed during July are assumed to be almost the same or very similar to those for the early part of the growing season.

### Foliar δ$^{13}$C values under different hydrological conditions
#### Foliar δ$^{13}$C values in normal dry-wet conditions

In the normal dry-wet SKA transect, in the absence of waterlogging during 2015, *S. boganidensis* grew at PA with more available water than those at PB or PC. The willow

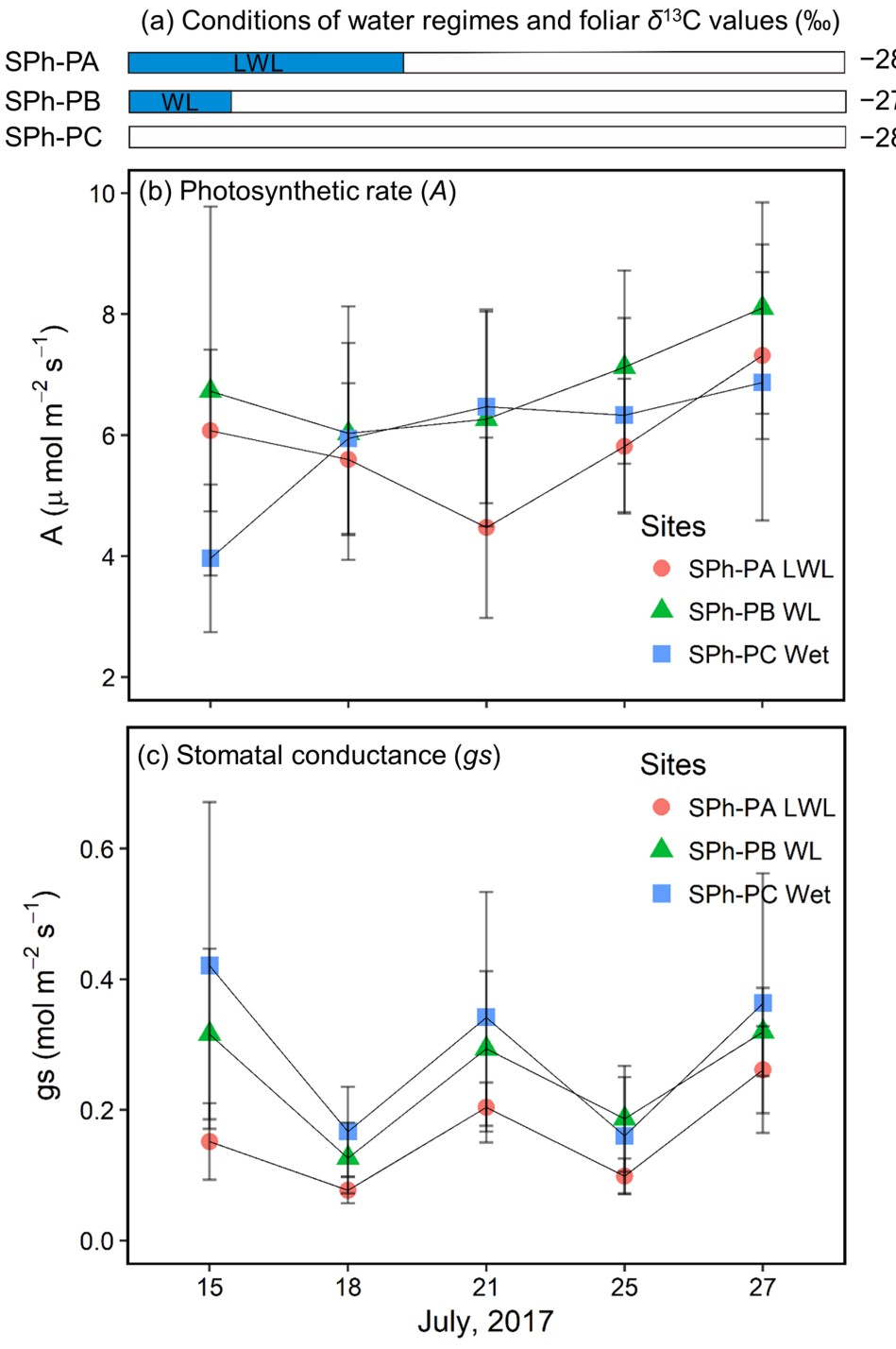

**Figure 6 Hydrological conditions in SPh, and results of physiology monitoring.** (A) Hydrological conditions in SPh-PA (circles), -PB (triangles), and -PC (squares), with the period of submergence shown in blue shaded bars and relative foliar $\delta^{13}C$ values (‰), and (B) photosynthetic rate ($A$, μmol m$^{-2}$s$^{-1}$) and (C) stomatal conductance ($gs$, mol m$^{-2}$s$^{-1}$) with fixed radiation (600 μmol m$^{-2}$s$^{-1}$), during July 13th–27th. Mean ± SD.

leaves at PA were largely depleted in $^{13}C$, with the difference of $\delta^{13}C$ value by about 2‰, relative to those at PB or PC (Fig. 3). These results are consistent with the well-known fact that, under dry conditions, low $gs$ results in high foliar $\delta^{13}C$ values; whereas, under wet conditions, high $gs$ leads to low foliar $\delta^{13}C$ values (Eq. (3)); all of which are consistent with the common findings with respect to $gs$ in numerous studies (*Chen, Bai & Han, 2002*; *Peri et al., 2012*; *Schifman et al., 2012*). The decrease in the foliar $\delta^{13}C$ value between 2015 and 2016 at PBs and PCs along transects SKA and SKB, and at PB on the transect SBoydom are thus, also attributable to stomatal regulation of gas exchange between dry and wet conditions.

### Foliar $\delta^{13}C$ values under sporadic waterlogging

Willow leaves at the PAs of SKB and SBoydom in 2015 have the $\delta^{13}C$ values similar to or higher than those in their respective PB, even though PAs and PBs were situated under wet and dry conditions, respectively. The high $\delta^{13}C$ values in PAs can be explained by stomatal closure under waterlogging conditions. Decreasing $gs$ was demonstrated under extremely wet conditions (*Gomes & Kozlowski, 1980*; *Olivella et al., 2000*; *Copolovici & Niinemets, 2010*; *Li & Sugimoto, 2017*). We suppose, as has been reported, that the flooding reduces root hydraulic conductance and thereby, leaf water potential (*Olivella et al., 2000*; *Islam & Macdonald, 2004*), which in turn leads to decreased $gs$. *Else et al. (2001)* also suggested that stomatal closure upon waterlogging is caused by the production of abscisic acid, which may be related to the decrease of root hydraulic conductance and leaf water potential. Moreover, in the present study, we observed a rapid recovery in $A$ but a slow recovery in $gs$ after waterlogging ended (Figs. 6B and 6C). Thus, when the waterlogging was short and continual, low $gs$ can contribute more to the foliar $\delta^{13}C$ values than $A$, resulting in the high foliar $\delta^{13}C$ values at SPh-PB, although the lack of statistical significance (Fig. 6A). The co-occurrence of low $gs$ with high foliar $\delta^{13}C$ values was also observed under waterlogging in field trials (*Ewe & Sternberg, 2003*; *Anderson et al., 2005*) as well as in pot experiments (*Li & Sugimoto, 2017*). Therefore, we suggest that $gs$ contributes more than $A$ to the foliar $\delta^{13}C$ values under sporadic waterlogging caused by medium flooding ("WL"; Fig. 3). The combined results of this and previous research indicates that increasing of the foliar $\delta^{13}C$ values (+0.8 ± 0.6‰) at SKA-PA between 2015 and 2016 were caused by stomatal closure in response to the waterlogging in 2016 ("WL"; Fig. 3). Very similar increases in the foliar $\delta^{13}C$ value were also observed in the SBoydom-PC, even though it was relatively distant from the river. The wetlands near SBoydom-PC may cause waterlogging similar to that experienced in the riverside sites (Fig. 3).

### Foliar $\delta^{13}C$ values under continuous long period waterlogging

Sporadic waterlogging ("WL"; Fig. 3) related to medium flooding increased the willow foliar $\delta^{13}C$ values. In 2016, however, the foliar $\delta^{13}C$ values at SKB-PA (waterlogging) were only slightly higher (approximately 0.4‰) than those at SKB-PB and SKB-PC. The foliar $\delta^{13}C$ values at SBoydom-PA (waterlogging) were even lower (approximately 1‰) than those at SBoydom-PB. Moreover, between 2015 and 2016 the foliar $\delta^{13}C$ values at PAs

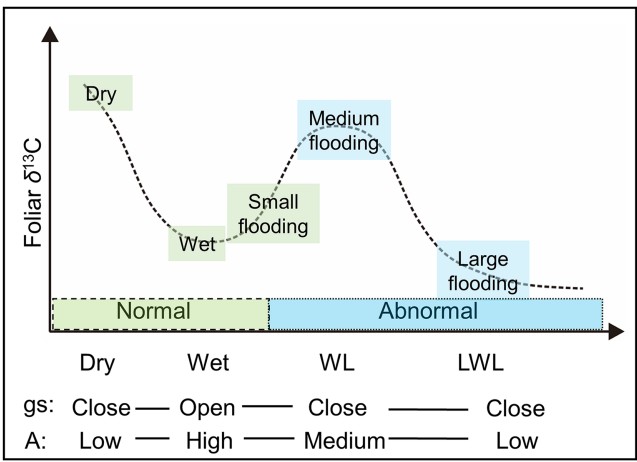

**Figure 7 The possible foliar δ¹³C values with respect to physiological responses to various hydrological conditions.** *gs*: stomatal conductance, *A*: photosynthesis activity. Dry and wet are without waterlogging, and WL and LWL represent waterlogging (continual) and long period waterlogging (continuous), respectively.

on SKB and SBoydom decreased by 1.7 ± 0.6‰ and 2.1 ± 0.9‰, respectively, despite waterlogging occurring in both years. To date, very few studies have investigated the reasons for the lack of changes or negative shifts in foliar δ¹³C value under waterlogging conditions. We propose that, under the long period waterlogging ("LWL"; Fig. 3) as caused by large flooding observed at PAs on SKB and SBoydom in 2016, the changes in the *A* are also important factors controlling the foliar δ¹³C values, besides *gs*. In the present study, low *A* was observed during submergence, before July 20 and more flooded SPh-PA (Fig. 6C). It has been suggested that waterlogging induces low carboxylation rate by reducing the amount or activity of Rubisco enzyme (*Vu & Yelenosky, 1992*; *Islam & Macdonald, 2004*). According to Eq. (3), the foliar δ¹³C values should be dependent on both *A* and *gs*; thus, low *A* may have caused the negative shifts in the foliar δ¹³C value in LWL compared to WL, as the foliar δ¹³C values at SPh-PA were slightly lighter than at SPh-PB, albeit insignificantly so (Fig. 6A). Therefore, the low foliar δ¹³C values observed in the willows at the PAs on SKB and SBoydom in 2016, can be explained by this continuously low photosynthetic activity under long period waterlogging caused by large flooding. The low δ¹³C values, *A*, and *gs* under large flooding were previously reported for a pot experiment involving the invasive wetland grass *Phalaris arundinacea* (*Waring & Maricle, 2012*). The aforementioned findings together suggest the hypothesis that long period waterlogging (or large flooding) significantly reduces the foliar δ¹³C values compared to sporadic waterlogging (or small flooding), and that the contribution of *A* and *gs* is highly dependent on the frequency and magnitude of waterlogging events.

Thus, the large differences (Fig. 4) in hydrological conditions (i.e., "Dry," "Wet," "WL," and "LWL"; Fig. 3) found in transects suggest that the possible foliar δ¹³C values can correspond to scenario 3 in Fig. 1 (i.e., reduced foliar δ¹³C values in long period waterlogging), which can be well interpreted by that both *A* and *gs* are affected by hydrological gradients (Fig. 7).

However, we note that scenario 3 in this study is a very simplified, schematic hypothesis, without quantitative meaning. There are other several potential factors for controlling the foliar $\delta^{13}C$ values, for example mesophyll conductance ($gm$) (*Evans et al., 1986*). As $gs$ and $gm$ were found tightly coupled (*Vrábl et al., 2009*), and both controlled the limitation of $CO_2$ diffusion. Therefore, in our field study of the determining factors of foliar $\delta^{13}C$ values, we mainly focused on $gs$ which can be directly monitored. Detailed changes in the $\delta^{13}C$ value particularly between and within the four hydrological conditions will be illustrated in further studies with determination of these potential factors, including but not limited to $gm$ changes under different water regimes, the quantitative meaning of $gs$ and $gm$ on foliar $\delta^{13}C$ values, and the features of $gs$ and $gm$ in species with different water tolerance.

## Spatial difference in the willow foliar $\delta^{13}C$ value

Linear Mixed Models analysis of the local scale random sampling indicated that only the willows in the dry larch forest were slightly but statistically enriched in $^{13}C$ ($F_{4,168} = 2.58$, $P = 0.039$), compared to the other conditions, in 2016. It is accepted that the foliar $\delta^{13}C$ values are higher in dry than in wet conditions, due to stomatal regulation. In contrast, there was no statistical difference in the foliar $\delta^{13}C$ value for either year or among the hydrological conditions tested here (flooding, flooded, on land, and wetland) (Fig. 5). The first three conditions were situated near the river and at different levels of flooding, whereas the fourth was never affected by flooding, although it was still abundant in water. These results are likely consistent with the lack of difference in the foliar $\delta^{13}C$ value among various hydrological conditions in mesic regions or periods, as reported previously (*Garten & Taylor, 1992*; *Alstad et al., 1999*). Nevertheless, relatively minor variations in the $\delta^{13}C$ value in willows growing under these hydrological conditions cannot be explained by the common dry-wet stomatal regulation theory mentioned above. On the other hand, our transect data in this study can explain why there was a slight variation in the foliar $\delta^{13}C$ value among random sampling sites in local scale in response to hydrological gradients as follows.

In 2015, the water level in the river was low. Consequently, the hydrological status of the willows growing nearest the river in the "Flooded" zone, ranged from slight flooding (similar to "Wet" in Figs. 3 and 4; i.e., "small flooding" in Fig. 7) to continual flooding (results in continual waterlogging, e.g., "WL" in Figs. 3 and 4; i.e., "medium flooding" in Fig. 7). Slight flooding near the wet condition zone caused the stomata to open and low foliar $\delta^{13}C$ values. In contrast, medium flooding resulted in stomatal closure and high foliar $\delta^{13}C$ values. Therefore, under the "Flooded" condition, the $\delta^{13}C$ values varied between low and high. This behavior resembles the positive-negative shift in the foliar $\delta^{13}C$ value for the "On land" zone under dry-wet conditions. The same interpretation applies to the foliar $\delta^{13}C$ values measured in the "Wetland" (Fig. 5).

In contrast, in 2016, the water level in the river was high. Therefore, the hydrological status of the willows growing nearest the river in the "Flooding" zone varied between continual flooding (results in continual waterlogging, e.g., "WL" in Figs. 3 and 4; i.e., "medium flooding" in Fig. 7) and continuous flooding (leads to long period waterlogging, e.g., "LWL" in Figs. 3 and 4; i.e., "large flooding" in Fig. 7). Large flooding

reduced both, $A$ and foliar $\delta^{13}C$ values. As was the case for 2015, the conditions in the "Flooded" and "Wetland" zones of 2016 ranged between slight and continual waterlogging (similar to "Wet" and "WL," respectively in Figs. 3 and 4), although only waterlogging in "Flooded" zones was caused by floods (i.e., "small flooding" and "medium flooding"; Fig. 7). Therefore, the foliar $\delta^{13}C$ values ranged between low and high in the "Flooding," "Flooded," and "Wetland" areas. These responses resemble the positive-negative shift in the foliar $\delta^{13}C$ value observed for "On land" under dry-wet conditions.

In previous studies (*Garten & Taylor, 1992*; *Alstad et al., 1999*), very small differences in the foliar $\delta^{13}C$ value of plants growing near rivers were detected among the diverse hydrological conditions, where waterlogging frequently occurred. These minor differences can also be explained by the physiological responses of willows related to the different hydrological conditions (Fig. 7). If the $\delta^{13}C$ values in other organs correlate with those determined by the leaves, then historical records of the wide swings in hydrological conditions could be reconstructed using the $\delta^{13}C$ records, such as those obtained from tree ring cellulose.

## CONCLUSIONS

To illustrate the effects of hydrological conditions on the $\delta^{13}C$ values in leaves, we measured the foliar $\delta^{13}C$ values of willows at three different points, along three transects near the Indigirka River, under several major hydrological conditions (Fig. 7). Under normal hydrological conditions, the foliar $\delta^{13}C$ values were lower under wet conditions (along rivers and/or during a wet year) than under dry conditions (far from the river and/or during a dry year), because the former conditions allowed for stomatal opening. On the other hand, under abnormal hydrological conditions, such as waterlogging, high foliar $\delta^{13}C$ values were found, because medium flooding induced stomatal closure. Moreover, long period waterlogging decreased foliar $\delta^{13}C$ value by reducing photosynthetic activity. Thus, there was a small variation in the foliar $\delta^{13}C$ value ($-31.6$ to $-25.7$‰) in the Indigirka River lowland, despite large diversity in the hydrological conditions (Fig. 5). These results demonstrate that the foliar $\delta^{13}C$ values reflect hydrological conditions even in mesic environments (Fig. 7). If the foliar $\delta^{13}C$ values correlate with those in other organs and tissues (such as tree-ring cellulose), they can be used to reconstruct the hydrological and vegetation changes that have occurred in mesic regions. We suggest that further clarifying of the effects of waterlogging on the foliar or tree ring $\delta^{13}C$ values in conducting laboratory experiments under controlled conditions and in field can be highly useful for better interpretation in the $\delta^{13}C$ values of plant products.

## ACKNOWLEDGEMENTS

We kindly thank Profs. A. Maximov, A. Kononov, and the other members of the IBPC and Ms. T. Stryukova and Mr. S. Ianygin of the Allikha Nature Protection Office for supporting our fieldwork near Chokurdakh, and Dr. S. Tei, Mr. R. Shingubara and S. Takano for their assistance in both field and labwork. We also thank Mss. Y. Hoshino, S. Nunohashi, and H. Kudo for their supports in labwork, and Dr. L. Chen for his helpful advice in paper writing.

### Funding

This work was supported by the China Scholarship Council (No. 201406180095); Japan Science and Technology Agency (Belmont Forum, project COPERA). The funders had no role in study design, data collection and analysis, decision to publish, or preparation of the manuscript.

### Grant Disclosures

The following grant information was disclosed by the authors:
China Scholarship Council: 201406180095.
Japan Science and Technology Agency (Belmont Forum, project COPERA).

### Competing Interests

The authors declare that they have no competing interests.

### Author Contributions

- Rong Fan conceived and designed the experiments, performed the experiments, analyzed the data, contributed reagents/materials/analysis tools, prepared figures and/or tables, authored or reviewed drafts of the paper, approved the final draft.
- Tomoki Morozumi contributed reagents/materials/analysis tools, approved the final draft, helped with sampling.
- Trofim C. Maximov approved the final draft, supported the field work.
- Atsuko Sugimoto conceived and designed the experiments, authored or reviewed drafts of the paper, approved the final draft, supported the field work.

### Field Study Permissions

The following information was supplied relating to field study approvals (i.e., approving body and any reference numbers):

Field experiments were approved by Institute for Biological Problems of Cryolithozone, Siberian Branch of Russian Academy of Science, Hokkaido University, and North-Eastern Federal University.

### Data Availability

The raw data are provided in the Supplemental Files.

### Supplemental Information

Supplemental information for this article can be found online at http://dx.doi.org/10.7717/peerj.5374#supplemental-information.

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
