# Peer review of "Effect of floods on the δ13C values in plant leaves: a study of willows in Northeastern Siberia"

_PeerJ, doi:10.7717/peerj.5374_

## Round 0.1 · original submission · Minor Revisions

Dear Dr Fan,

We have received two favorable reviews for your submission. I agree with reviewers that the work you present is original and deserves publication. The reviewers point at some weaknesses that need your detailed attention. In addition, before the manuscript is accepted for publication you will need some additional concerns after my reading of the manuscript.

I think the authors did a good job at explaining the transect for the water logging treatments. However, isotope analyses were done on whole leaves, and the bulk of their biomass was made during the first couple of weeks after emergence. Analyses of whole leaves points to two potential shortcomings:

1- How an isotopic composition set during conditions experienced by trees for the early part of the month of June matches the gas exchange parameters measured in July? The authors do not mention this potential mismatch, or turnover of leaf biomass as a way to reconcile time integrated with spot measurements. This aspect needs to be added to the discussion or, alternatively, starch/sugar data added to the paper.

2- Likewise, if waterlogged and dry/wet conditions are seasonal it is important to discuss the effects of hydrology in the context of when leaves are formed and when most of the delta-13C of the leaves is set. Again this needs to be added to the discussion as alternative hypotheses for their data are possible.

The authors refer to the linkage between 13C and water use efficiency. The authors need to incorporate the role of mesophyll conductance in deviating 13C values from water use. The relationship between 13C of plant tissues and water use efficiency is not linear, as previously thought, and a section on the discussion about gm seems warranted as changes in gm can help explain the results at least partially. You could also calculate gm from A/Ci curves if you have them or even 13C of photosynthate of leaf if one assumes gm is constant. See Journal of Experimental Botany, Vol. 60, No. 8, pp. 2315–2323, 2009

I hope the authors can adequately respond to editor’ and reviewers’ concerns. Please contact me if you have any questions.

Reviewer 1 ·

Basic reporting

I recognize that this manuscript has a standard level of basic reporting.

Experimental design

I recognize that this manuscript has a very nice experimental design, although I feel that data set in this manuscript are relatively scarce or just minimum to conduct this study.

Validity of the findings

I recognize that the findings of this study is valuable and contributable to advances in the study of the related fields.

Additional comments

I am impressed with this manuscript. I recognized that this is a nice study, the demonstration here are valuable in that they impact a very general issue how we can interpret the hydrogel dynamics of the stable carbon isotopic composition of plant products, which is essential for accurate understanding of the ecology and evolution of plant vegetation in modern environments as well as in environmental changes from past to modern. To my knowledge, this manuscript will be the first publication, successfully to interpret the traditional hypothesis on the paradox frequently observed in plant carbon isotopic composition in wet-dry gradients.
As described in this manuscript, we have long studied the physiological response of plants to hydrogel dynamics, particularly for wet-dry gradients, with stable carbon, hydrogen, and oxygen isotopic compositions of plant products (cellulose, lipids, etc). This is because many scientists have simplistically believed that change in the stable isotopes of plant products is a single function of wet-dry gradients, so they have long neglected the other major factor, i.e., in addition to stoma closeness due to wet-dry gradients, photosynthetic activity can also influence much the change in the carbon isotope ratios of plants. It is also true that many scientists have well misunderstood, as hydrogen or oxygen isotope analysis of leaf water or plant cellulose is a unique solution for seeing the effect of stoma closeness on the carbon isotopes, independent of the degree of photosynthetic activity. However, in addition to increased in D/H or 18O/16O ratios due to evapotranspiration (which is closely related to wet-dry gradients), diversity in the water sources can also influence much the change in the oxygen isotope ratios of plants. So, to my knowledge, no study has long been conducted to see both effects of stoma closeness and photosynthetic activity in hydrogel dynamics. I thus recognize that this study used a very nice sample set successfully to see that plant carbon isotopic composition well responds diverse hydrological pressures, with respect to wet-dry gradients and its associated change of photosynthetic activity.
I also feel that the manuscript is well organized into a readable story, therefore I suggest that this manuscript should be publishable in PeerJ. I am really looking forward to seeing the published paper of this study very soon.

Reviewer 2 ·

Basic reporting

no comment

Experimental design

no comment

Validity of the findings

no comment

Additional comments

To Authors

This manuscript evaluated factors controlling the δ13C values of willow leaves in natural environments, especially under flooding and waterlogging conditions. I am really impressed this cutting-edge study.

It is true that we have usually interpreted the change in the δ13C values of plant leaves as the results of the block of CO2 input through stomata in dry conditions, compared to no-block in wet conditions. However, it is also very true that this interpretation should work only in a lot of assumptions, as the Authors well explained them in this manuscript. We thus have long suffered with ‘paradox’ in the interpretation of the δ13C values of plant leaves. Again, I recognized that this manuscript is nicely organized and well explain this ‘paradox’ on the change in the δ13C values of plant leaves, on the standard equation with balance between the CO2 input through stomata and the CO2 consumption attributable to photosynthesis activity. Moreover, this explanation is well verified with the new data from willow leaves under flooding and waterlogging conditions.
Although it is difficult that the most suitable place/samples for such experiments is found in nature environment, the Authors nicely found a riverside area where frequently floods and waterlogged, and leaves of willows, which are widely distribution around river, are likely record such various hydrological conditions. In Introduction, the Authors designed nice explanation of the relationship among the δ13C values, stomata conductance, and photosynthesis activity in plant leaves, which will be highly useful for readers of this manuscript for understanding that among these three factors in plant leaves.
I feel thus that this manuscript has a nice flow with high readability of all contents from Introduction to Discussion.
I can also recognize that the new findings in this study is essential for applying and interpreting of stable carbon isotopic composition of plant products, even in pot experiments as well as environments specific to floodplain with frequent change of water levels. Moreover, I think that advances in understanding the physiological mean on the isotopic compositions (including not only carbon but also other elements) of plant leaves in this study can contribute much how we can see the grow system, ecological function, and biosynthesis/metabolism of plants in future studies.
Based on these thought, I can recommend this manuscript to PeerJ, as a nice publication in there. However, I also feel that the Authors should be carefully reconsider three unclear points before publication (see specific comments below).


Specific comments:

1. “Balance between A and gs” is the key effect on the δ13C values of plant leaves, for not only “LWL” but also “WL”. However, in the “WL” section, only gs is emphasized as the almost solo factor for increasing the δ13C values, but is neglecting the view of “balance between A and gs”. I thus worry about that such explanation leads readers toward misunderstanding of this key point. I suggest that the Authors modify the corresponding sentences/paragraph(s) accordingly to the “balance” that the gs has a much contribution compared to A in the case of plants under the “WL” conditions.

2. Can you show the Figure 4 with the data of 2015 or 2016?
The δ13C data of plant leaves were found in willows collected from 2015 and 2016, but the A and gs data were collected in 2017. Also, although I can understand, many readers who are not familiar to such plant δ13C studies may have a common question why the Authors did not show the row values (but relative values to T=0 or PC) for the A and gs in Figure. 4.
I think that this manuscript already did sufficient explanation for the point of A/gs balance, even if the Author did not show Figure 4. However, on the other hand, I very much agree with the Authors’ intention that such physiological data, even in 2017, is important to see the physiology changes of willows under various water conditions in the studied area. Thus I suggest that the Figure 4 move to Appendix from Discussion, to enhance readability of this manuscript and to focus the changes in the δ13C value of plant leaves.
Also, please re-check carefully the following points in Figure 4:
> The length of blue bars, especially the final date of flooding, on each of conditions.
> Can the Authors show the specific hydrological condition (e.g., “WL”) for plants measured in Figure 4?

3. I think that the values for mean, cap-delta, standard deviation, etc., reported in this manuscript may be miscalculated for several places. Please re-check carefully for all.


L38: ca, use Italic, Subscript letter for “a”

L54-57: gs and gc, what is difference gs from gc?

L102: predominate

L137: Diameter

L152: 20 m

L163, 267, Table A2, and Figure 1(a): There are 29 sits in text, but 28 sites in Table A2. Also I cannot find 29 sites in Figure 1(a).

L181: S. boganidensis

L214-215: Use italic for S. richardsonii, S. glauca, and S. boganidensis

L237: SBoydom

L250-251: Figure 5

L257: -0.5‰

L309: SBoydom

L322: July 20 and more

L323: Figure 4b

L353: the δ13C

Fig.3: means±SD

Figs. 5 and 6: Please explain “a”, “b”, and “ab”

---

## Round 0.2 · Minor Revisions

Thanks for submitting your revised manuscript. You have addressed the concerns in the text satisfactorily. Therefore I will recommend acceptance once the comments below are incorporated. I still found a large number of grammatical errors. I went through the most significant ones to expedite the process of your manuscript. Please understand the use proper English in submitted manuscripts is the responsibility of the authors. Thanks for submitting your work to PeerJ.

Editorial comments:

L 26 - delete "in the present study"
L 27 - varIED,
L 27 - under the "different hydrological regimes,..."
L 28 - delete "the following processes"
L 30 - delete "however"
L 31 - delete "moreover"
L 36 - regionS.
L 40 - Pee Dee not Peedee
L 41 - try this " to estimate changes in carbon flux as plant physiology responds to environmental change"
L 44 - add (∆13C) after carbon isotopic fractionation
L 83 - delete "a' before "drought" or change conditions (plural) to condition (singular)
L 87 - delete "and" before "Lin"
L 102 - than under low gs, A reduction of A...
L 115 - indicate region, country for Indigirka River
L 117 - capitalize River
L 117-119 - Try this "Thus, because willows in the Indigirka River lowland are exposed to an increase frequency of river floods and have a higher chance of being submerged, they are a good candidate species to study the effects of flooding on the δ13Cp values of leaves in relation to A and gs. "
L 120 - delete "therefore"
L 122 - try "long periodS OF waterlogging" and delete "conditions" before (Figure 1)
L 123 - try "under theSE flooding REGIMES in the"
L 124 - capitalize NorthEastern
L 144-146 - I do not understand what this means? Also one sentence cannot be used to make a paragraph.
L 159 - delete "around"
L 161 - " consistent with A NDVI study..."
L 161 - Annual new ABOVEGROUND PRIMARY production...
L 164 - NAPP should be substituted for the common name "ANPP" which stands for Aboveground Net Primary Production
L 164-165 - Move this sentence to results as results should not be given in methods section
L 191 - Pee Dee not Peedee
L 194 - delete "in the single site"
L 195 - change ranged for RANGING
L 198 - change supportive for supporting and delete "of our discussion"
L 214 - try this "Five measurements were taken every 2-3 days between ..."
L 220 - change "meanwhile" with "with"
L 251 - substitute "points" for "sample locations"
L 264 - growing seasons OF 2015 and 2016...
L 280 - Try this "In the summer of 2017, TREES AT THE Sph-PA SITE wERE continuously SUBMERGED until July 20;..."
L 282 - change "points" for "locations"
L 283 - delete " only proximately"
L 290-292 - This is written as discussion. Move or delete
L 292 - it should be WERE not WAS
L 305 - add error after 0.6 per mil
L 308 - change "entire" for "early part of the"
L 313 - add "isotope values" after 13C or use delta notation.
L 333 - add a comma (,) after "Thus"
L 339 - gs abbreviation already been defined so delete "stomatal conductance"
L 388 - which potential factors? please list
L 420 - As was the case FOR 2015,
L 448 - remove "and vegetation" as this is a stretch
L 451 - I think you mean "interpretation" rather than "application"

---

## Round 0.3 · accepted · Accept

Thanks for sending the revised manuscript. There are still a small number of typos and grammar issues that I hope you can correct during the production stage.

#